

# The impact of hsa-miR-1972 on the expression of von Willebrand factor in breast cancer progression regulation

Changjiang Yu[1], Tao Zhang[2], Fan Chen[1] and Zhenyang Yu[1]

[1] Department of Breast and Thyroid Surgery, Maternal and Child Health Care Hospital, Chongqing, China
[2] Department of Breast and Thyroid Surgery, Chongqing Jiangjin District Central Hospital, Chongqing, China

## ABSTRACT

**Background:** Breast cancer (BC) is one of most frequent female malignancies that poses multiple challenges in treatment and prevention. This study aimed to explore the role of miRNAs and their target genes during the BC progression.

**Methods:** Based on the BC data (113 normal and 1,118 tumor samples) from the TCGA-BRCA dataset, a single-sample gene set enrichment analysis (ssGSEA) was applied to calculate the cancer migration scores, and weighted gene co-expression network analysis (WGCNA) were performed using the WGCNA R package, with a focus on the set of genes associated with cancer migration. Key modules and hub genes related to cell migration and signaling pathways were identified. Survival analysis of hub genes was conducted using the survminer R package, and prediction of regulatory miRNAs were performed to analyze their impact on BC prognosis. In addition, the BC cell lines MCF-7 and MDA-MB-231 were used to further explore the effect of hsa-miR-1972 mimics on the gene expression and angiogenic factor regulation.

**Results:** The study classified important modules (MEblue, MEmagenta, MEpink, and MEfloralwhite) associated with cell migration and identified three hub genes, namely, MRPL20, COL4A1 and VWF. Survival analysis showed that certain hub genes with a low expression were related to a poor prognosis, whereas low-expressed COL4A1 and VWF were related to better survival outcomes. We also found that hsa-miR-1972 mimics significantly downregulated critical genes involved in BC metastasis and angiogenesis and effectively inhibited the proliferation of BC cell lines, showing a strong therapeutic potential. Manipulation of VWF expression in cells overexpressing hsa-miR-1972 had significant effects on the malignant markers and angiogenic factors, suggesting a novel therapeutic direction for BC treatment.

**Conclusion:** Our study highlighted the complex interplay of genetic factors in BC progression as well as the therapeutic potential of targeting specific miRNAs and their related hub genes. These findings provided novel insights into the pathogenesis of BC and suggested new direction for the therapeutic development for the cancer.

Corresponding authors
Changjiang Yu,
13883270323@163.com
Zhenyang Yu,
18223159797@163.com

## INTRODUCTION

Breast cancer (BC) is one of most frequent female malignancies with high incidence and mortality rates (*Yin et al., 2023*). In 2020, there were about 2.3 million new cases and 685,000 deaths worldwide (*Chou et al., 2024*). BC exhibited higher heterogeneity, according to the expression of human epidermal growth factor receptor 2 (HER2), progesterone receptor (PR), and estrogen receptor (ER), BC could be categorized into four main molecular subtypes of triple-negative breast cancer (TNBC), HER2-positive, Luminal B, and Luminal A (*Fumagalli & Barberis, 2021*). Each subtype of BC is treated with different strategies, including endocrine therapy, anti-HER2 therapy, chemotherapy, radiotherapy, and mastectomy (*Tan et al., 2023*). However, all the treatments can cause side effects and patients may develop resistance to these drugs. In particular, there is no standardized treatment protocol for TNBC, which also increases the complexity and difficulty of clinical treatment for the cancer (*Burguin, Diorio & Durocher, 2021*; *Ala Eddine-Le Jallé, Aubert & Nos, 2013*). Overall, treatment resistance, personalization of treatment regimens and the selection of effective treatment options for different TNBC patients are currently the major challenges to BC treatment and prevention. Given the heterogeneity, some novel precise strategies of BC treatment were emerged, such as immunotherapy (*Hassan et al., 2024*), targeted drugs (Herceptin), targeting cell interactions, and immune checkpoint inhibitors (*Guo et al., 2023*). Additionally, cell migration is crucial for the invasion and metastasis of cancer and collective cell migration is deemed to be the main migration mode, which is relevant to poor prognosis of BC patients (*Hapach et al., 2023*). The heterogeneity of BC migration and metastasis phenotypes may be a challenge for BC patients to determine appropriate treatment methods (*Choi et al., 2024*). MRTF-A was reported to regulate migration-related genes by integrating the Rho-ROCK-actin and Wnt-β-catenin pathways, which could provoke BC cell migration (*He et al., 2018*). Therefore, it is necessary to better understand the role of BC migration phenotype and its related genes in the progression of BC.

The roles of microRNAs (miRNAs), as disease biomarkers, in the chemoresistance, prediction of systemic therapeutic response, and novel miRNA-based drug delivery strategies in cancers have received much attention from the field of BC research (*Chong, Yeap & Ho, 2021*). Previous study has shown that miRNAs play key roles in the development, maintenance, metastasis, and chemoresistance of BC (*Singh et al., 2023*). For instance, the level of miR-222-3p in high-metastatic MDA-MB-231 cell line was higher than that in low-metastatic MCF-7 cell line, and the inhibition of endogenous miR-222-3p expression in MDA-MB-231 reduced the proliferation of 20–40% and migration of ~30%, indicating that miR-222-3p partially regulated the invasive phenotype of MDA-MB-231 cell line (*Phannasil, Akekawatchai & Jitrapakdee, 2023*). Upregulated miR-34a is associated with docetaxel resistance, whereas inhibiting miR-34a improves patients' responsiveness to docetaxel (*Kastl, Brown & Schofield, 2012*; *Kassem et al., 2019*). In addition, the miR-200 family is considered as a key regulator of tumor development as they can inhibit cancer invasion and metastasis through suppressing key inducers of

epithelial-mesenchymal transition (EMT) such as ZEB1, ZEB2, and SLUG (*Mutlu et al., 2016*; *Zhang, Xu & Li, 2014*). Moreover, miRNAs have also been investigated for their roles in neoadjuvant chemotherapy. Past research demonstrated that miR-638 (*Di Cosimo et al., 2019*) and a low level of miR-451a are associated with chemoresistance, whereas higher expressions of miR-200c-3p, miR-23a-3p and miR-214-3p contribute to chemoresistance (*Xing et al., 2021*). These findings provided a new strategy for personalized medicine to predict chemotherapy response by monitoring the expression patterns of miRNAs in cancer patients. Polymer-based carriers such as polyethyleneimine (PEI) and poly lactic-co-glycolic acid (PLGA) have been extensively investigated for miRNAs and related drug delivery. Previous study developed anti-miR-155 polymer complexes using cross-linked polyethyleneimine (PEI-SS) and demonstrated the potential to effectively inhibit BC growth *in vivo* (*Li et al., 2021*).

Besides, hsa-miR-1972 was considered crucial in the etiology of larynx cancer (*Ekmekci et al., 2019*). Cannabidiol could specifically alter the expression of miRNA and played a potential role in inducing apoptosis in neuroblastoma cells by downregulating hsa-let-7a and upregulating hsa-miR-1972 (*Alharris et al., 2019*). However, the research on hsa-miR-1972 in the BC progression has not been reported. Although miRNAs have shown great potential in BC therapy, their low levels in the bloodstream make the identification difficult using conventional methods. To improve the accuracy and sensitivity of detecting miRNAs, global expression analysis and multiple technological platforms for miRNA assessment, such as next-generation sequencing (NGS) and reverse transcription quantitative PCR (RT-qPCR), have also been widely applied. Apart from the potential in circulating miRNAs as BC biomarkers, differences in patients selected and experimental methods may also cause discrepancies in results, suggesting the need to further analyze the role of miRNAs (*Cardinali et al., 2022*).

This study delved into the complexity of BC progression, with a focus on the role of miRNAs and their target genes during the cancer progression. The interactions between VWF and hsa-miR-1972 in influencing BC progression was explored. The current findings indicated that inhibiting the expression of VWF and overexpressing hsa-miR-1972 significantly affected the malignant tumor markers and angiogenic factors, which could be developed as a novel strategy for treating BC patients.

## MATERIALS AND METHODS

### Data sources
The sequencing dataset for the breast invasive carcinoma project (TCGA-BRCA), which contained 113 normal samples and 1,118 tumor samples, were downloaded from The Cancer Genome Atlas (TCGA, https://portal.gdc.cancer.gov/). The annotation and clinical information of the samples were acquired based on the rowData and colData functions in the TCGAbiolinks package (*Colaprico et al., 2016*) (parameters: data.category = "transcriptome analysis", data.type = "gene expression quantification", and workflow. type = "STAR - Count"). Subsequently, annotations and clinical data related to the samples were extracted using the rowData and colData functions.

## Cancer migration scores

In order to identify the genes associated with BC cell migration, the cancer migration scores were calculated by ssGSEA (*van Roosmalen et al., 2015*) based on the set of cancer migration-associated genes from collected from *van Roosmalen et al. (2015)* The expression matrix of for these gene sets in each sample in the TCGA-BRCA expression was transformed by log2 (FPKM+1) to calculate the enrichment scores.

## Gene co-expression network development

For screening the critical modules related to BC cell migration, weighted gene co-expression networks were developed based on TCGA-BRCA gene expression data using the WGCNA package (*Langfelder & Horvath, 2008*). The expression matrix was preprocessed using the log2 (FPKM + 1) transform and then the outliers were removed by hierarchical clustering. The similarity matrix was then constructed using Pearson correlation and transformed to an adjacency matrix using a power function to ensure a scale-free nature of the network. The optimal soft-threshold power $\beta$ was determined to satisfy the scale-free topology criterion (correlation of log (k) with p(k) $\geq$ 0.85) to convert adjacency matrix into a topological overlap matrix (TOM). Hierarchical clustering of highly similar gene modules was performed, followed by using Pearson correlation analysis to identify modules with the highest correlation to BC cell migration. The clusterProfiler package (version 4.8.2) was used to analyze the biological functions of genes within the target modules with the parameters of keyType = "ENTREZID", *p*valueCutoff = 0.05 and qvalueCutoff = 0.1 (*Yu et al., 2012*; *Song et al., 2023*).

## Development of protein-protein interaction network and identification of hub genes

The key hub genes were identified by protein-protein interaction (PPI) network. After excluding disconnected nodes, genes from each module were entered into the STRING database (https://string-db.org/) (confidence > 0.4) to construct a PPI network (*Szklarczyk et al., 2023*). MCODE algorithm and cytoHubba in the Cytoscape (version 3.8.0) were used to for cluster identification and calculating the network connectivity. The top 10 genes showing the highest connectivity were defined as hub genes (*Shannon et al., 2003*).

## Validation of hub genes

Association of hub genes with clinical outcomes and prognosis of BC was assessed by Kaplan-Meier (KM) survival analysis. Based on the optimal cutoff value determined by the surviviner package, patients were divided into high- and low-expression groups and survival curves were plotted using the survminer package (*Kassambara et al., 2017*).

## Prediction of miRNAs regulating the hub genes

The miRNAs that targeted identified hub genes were predicted based on three databases, including miRDB (https://mirdb.org/), miRanda (http://www.microRNA.org), and TargetScan (https://www.targetscan.org/vert_80/) were used to predict miRNAs. The

results from the three databases were integrated to determine the miRNA-target gene pairs.

## Cell culture and transfection

DMEM (Gibco, Grand Island, NY, USA) supplemented with fetal bovine serum (Gibco, Grand Island, NY, USA) and penicillin/streptomycin was used to culture human BC cell lines (MDA-MB-231 and MCF-7) in 5% $CO_2$ at 37 °C. The two cell lines were purchased from Typical Culture Reserve Center of China (Shanghai, China). The si-RNA and oe-RNA of VWF (Sangon, China) and negative control as well as miR-1972 mimic or its control were transfected into the cells using Lipofectamine 2000 (Invitrogen, Waltham, MA, USA). See Table 1 for the sequences of small interfering RNAs.

## Cell viability

The effect of hsa-miR-1972 mimic on the growth of BC cells (MDA-MB-231 and MCF-7) was assessed by cell viability assay. The cells ($1 \times 10^3$ cells) with different treatments were cultured in each well of 96-well plates. Cell Counting Kit-8 (CCK-8) solution (Beyotime, Jiangsu, China) was applied at specific time points. After cell incubation at 37 °C for 2 h (h), the $OD_{450}$ value was detected using a microplate reader (Thermo Fisher, Waltham, MA, USA).

## Western blot

Western blot assay was applied to evaluate the protein expression of VWF in BC cells. Total protein was isolated using RIPA buffer and then quantified. Protein sample was separated by 10% SDS-PAGE, then moved onto a PVDF membrane (0.45-μm) and blocked with 5% skimmed milk for 2 h. The corresponding primary antibodies (anti-VWF (11778-1-AP, 1/1,000) and anti-GAPDH (ab8245, 1/5,000)) were incubated with the membrane at 4 °C overnight. The secondary antibody was incubated with the protein at room temperature for 1 h. Immunoblots were developed using an ECL fluorescence detection kit (Beyotime, Shanghai, Jiangsu, China) and visualized with a Tanon 4,600 system (Tanon Science and Technology Co., Ltd., Shanghai, China).

## QRT-PCR

Total RNA from MCF-7 and MDA-MB-231 cell lines was extracted applying TRIzol (Thermo Fisher, Waltham, MA, USA) reagent. The cDNA was extracted from 500 ng of RNA with HiScript II SuperMix (Vazyme, Nanjing, China). PCR amplification was performed under conditions of 10 min at 94 °C, 10 seconds (s) at 94 °C and 45 s at 60 °C. GAPDH served as an internal reference. Table 2 displayed the sequence list of primer pairs for the targeted genes. The relative expression levels were calculated with the $2^{-\Delta\Delta CT}$ method (*Sindhuja, Amuthalakshmi & Nalini, 2022*).

## Dual-luciferase reporter assay

The specificity of the interaction between hsa-miR-1972 mimic and VWF was detected by the dual-luciferase reporter assay. The coding sequence (CDS) of VWF containing mutant or wild-type (wt) was ligated into the firefly luciferase expression vector pMIR-REPORT

**Table 1 Sequences for transfection.**

| Gene | Target sequence (5′–3′) |
| --- | --- |
| si VWF | CTCATTTGCAGGGGAAGATGATT |
| miR-1972 mimics | UCAGGCCAGGCACAGUGGCUCA |

**Table 2 Primer sequences targeting genes.**

| Gene | Forward primer sequence (5′–3′) | Reverse primer sequence (5′–3′) |
| --- | --- | --- |
| CDH1 | GCCTCCTGAAAAGAGAGTGGAAG | TGGCAGTGTCTCTCCAAATCCG |
| ITGB8 | CTGTTTGCAGTGGTCGAGGAGT | TGCCTGCTTCACACTCTCCATG |
| MAP2K1 | GTGGTGAGTTTCATAGCCAGCAG | AGAAACTTCCACAACTTGTCTCAG |
| MMP9 | GCCACTACTGTGCCTTTGAGTC | CCCTCAGAGAATCGCCAGTACT |
| NEK1 | GGACAGTATGAACATTACCATGCC | CTGGACGAACTCCAGGCAGAAT |
| SHC1 | ACAGCCGAGTATGTCGCCTATG | CAATGGTGCTGATGACATCCTGG |
| GAPDH | GTCTCCTCTGACTTCAACAGCG | ACCACCCTGTTGCTGTAGCCAA |

(Sangon, Shanghai, China). Transfection of VWF-mut and VWF-wt was conducted applying Lipofectamine 3000 (Invitrogen, Carlsbad, CA, USA) together with miR-NC with miR-mimic (Sangon, Shanghai, China). The pTK-Renilla luciferase reporter (Genomeditech, Inc., Shanghai, China) containing the full-length Renilla luciferase (which was used to normalize the luciferase activity) was co-transfected into the cells and the fluorescence intensity was quantified 48 h after transfection with Dual-Luciferase Assay System (Promega, Inc., Madison, Wisconsin, US).

## Transwell

Transwell assay was performed to evaluate the cell migration ability of MDA-MB-231 and MCF-7. The cells ($5 \times 10^4$) were inoculated into the upper chamber of the Transwell added with serum-free medium, while complete DMEM medium were added to the lower layer. After incubation for 24 h, the cells were fixed using paraformaldehyde and dyed by 0.1% crystal violet.

## Enzyme-linked immunosorbent assay

The concentrations of angiogenic factors, VEGF, bFGF, TGF-β, and HIF-1α, were detected using enzyme-linked immunosorbent assay (ELISA) kits following the manufacturer's instructions (Elabscience). The total protein concentration of each sample was then determined using the Bradford assay (Cat. ab119216; Abcam, Cambridge, UK) and concentrations were shown in the form of pg/mL.

## Statistical analysis

All measurement data were represented as mean ± standard deviation and all experiments were performed in independent triplicates. Student's t-test was used to compare differences in continuous variables between two groups, and all the computational analyses
were conducted in R (version 4.3.1). GraphPad Prism (version 8.0.2) was applied for the statistical analysis. A $p < 0.05$ was considered statistically significant.

## RESULTS

### WGCNA analysis and identification of the hub genes

The ssGSEA and WGCNA were conducted based on the cancer migration-related gene set in the samples in TCGA-BRCA, noticeably, the MEblue module was positively correlated with cell migration, the MEmagenta and MEpink modules were correlated with cell migration enhancement and facilitation, and the MEfloralwhite module was negatively correlated impaired cell migration (Fig. S1). To further explore the functional roles of these modules in BC development, enrichment analysis was performed to analyze their biological processes. It was found that the genes in the MEblue module were mainly involved in mRNA processing, autophagy and TOR signaling (Fig. 1A), whereas genes in the MEmagenta and MEpink modules were mainly enriched in biological processes involving the migration of multiple cell types (Figs. 1B, 1C). In addition, a PPI network was developed based on the genes in each module to further screen key hub genes. Specifically, MRPL20, TUFM, MRPL12, MRPL43, MRPL21, MRPL41, AURKAIP1NDUFB7, MRPS34, and ATP5F1D were the hub genes in the MEblue module (Fig. 1D); MRPL13, POLR2K, COPS5, ELOC, NUDCD1, TATDN1, ENY2, DCAF13, MTERF3, and UBE2V2 were the hub genes in the MEfloralwhite module (Fig. 1E); and COL5A1, COL4A1, COL6A2, COL6A1, BGN, COL6A3, FBN1, POSTN, DCN, and LUM were the hub genes in the MEmagenta module (Fig. 1F); KDR, PECAM1, CDH5, FLT4, VWF, TIE1, GAPDH, FGF7, SOX18, and GJA4 were the hub genes in the MEpink module (Fig. 1G). Taken together, our study revealed the expression patterns of the set of genes related to BC cell migration and their potential functions in the cancer progression, especially in cell migration and signaling pathways.

### Hub genes associated with BC prognosis

To further explore the impact of the hub genes on the prognosis of BC samples in the TCGA-BRCA cohort, we performed survival analysis for all the hub genes. KM survival curves showed that low-expressed MRPL20, MRPL12, AURKAIP1, NDUFB7, ATP5F1D, and BGN were in BC patients were correlated with a poor survival trend, whereas low-expressed COL4A1 and VWF were related to a significant survival advantage (Figs. 2A–2H).

### Prediction of miRNAs regulating the hub genes

The miRDB, miRanda, and TargetScan databases were used to predict the miRNAs that targeted the key hub genes MRPL20, MRPL12, AURKAIP1, NDUFB7, ATP5F1D, BGN, COL4A1, and VWF. By combining the prediction results from the three databases, we obtained 51 intersections of miRNAs with target gene pairs (Table S1, Fig. 3A). Detailed information of these miRNA-target gene pairs was shown in Fig. 3B. Among these hub genes, COL4A1 is responsible for encoding the α1 chain of type IV collagen, which is a key

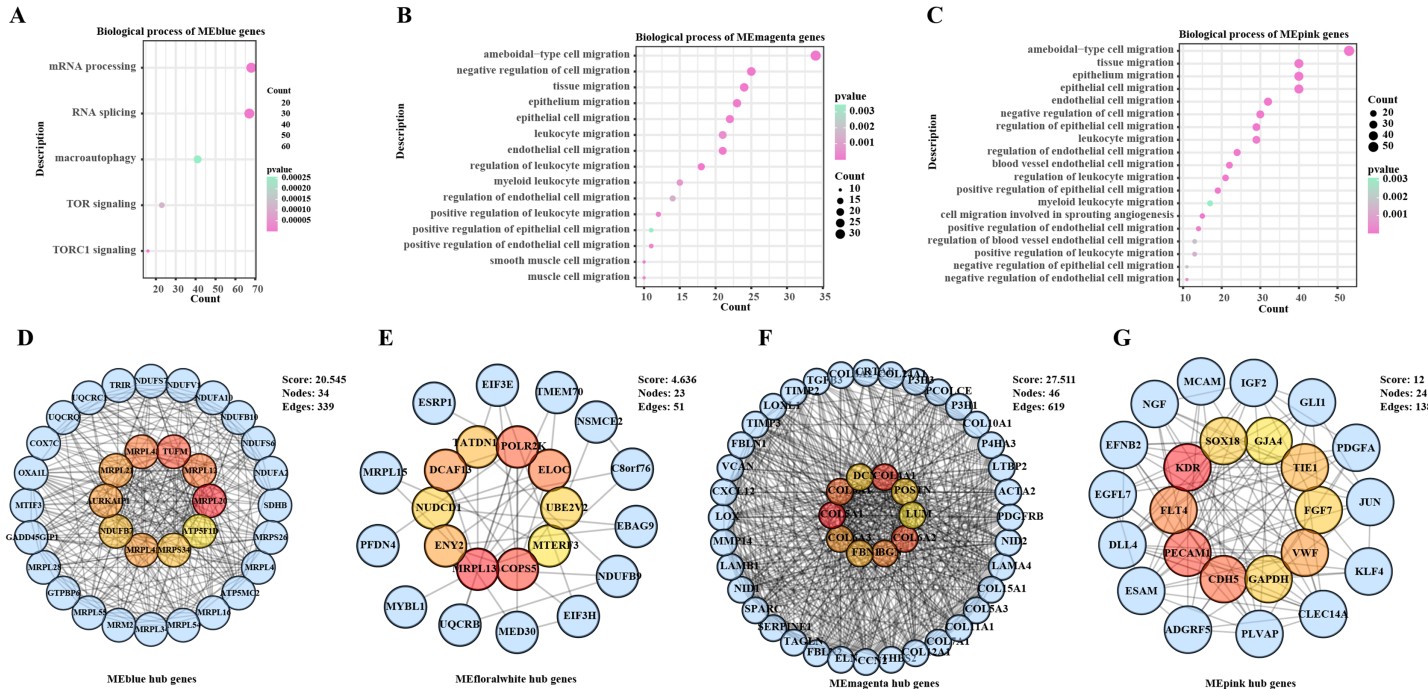

**Figure 1 Biological functions and PPI networks of genes within the module.** (A) Biological process enrichment analysis of MEblue module genes. (B) Biological process enrichment analysis of MEmagenta module genes. (C) Biological process enrichment analysis of MEpink module genes. (D) Identification of hub genes of genes in MEblue module by PPI network. (E) Identification of hub genes of genes in MEfloralwhite module by PPI network. (F) Identification of hub genes of genes in MEmagenta module *via* PPI network. (G) Identification of hub genes of genes in MEpink module *via* PPI network.

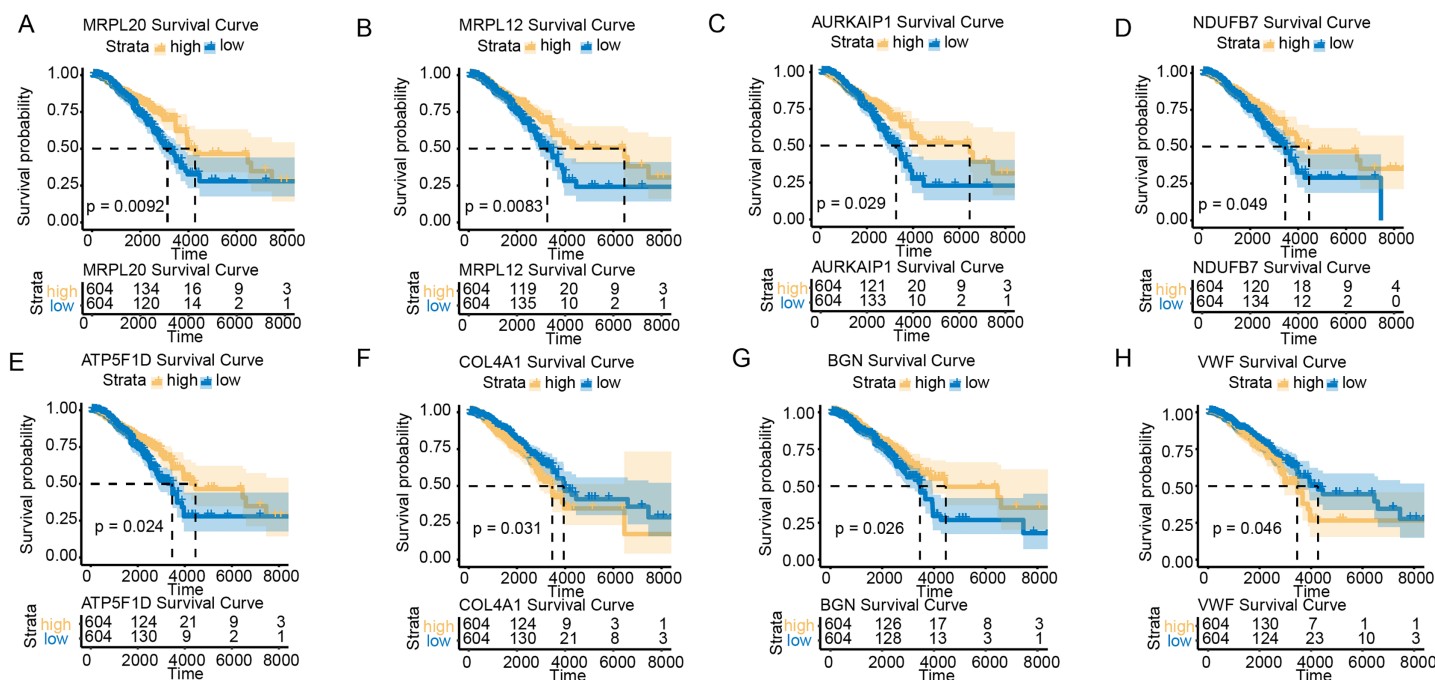

**Figure 2 Survival analysis of hub genes in BC patients.** (A) K–M survival curve of MRPL20. (B) K–M survival curve of MRPL12. (C) K–M survival curve of AURKAIP1. (D) K–M survival curve of NDUFB7. (E) K–M survival curve of ATP5F1D. (F) K–M survival curve of COL4A1. (G) K–M survival curve of BGN. (H) K–M survival curve of VWF.

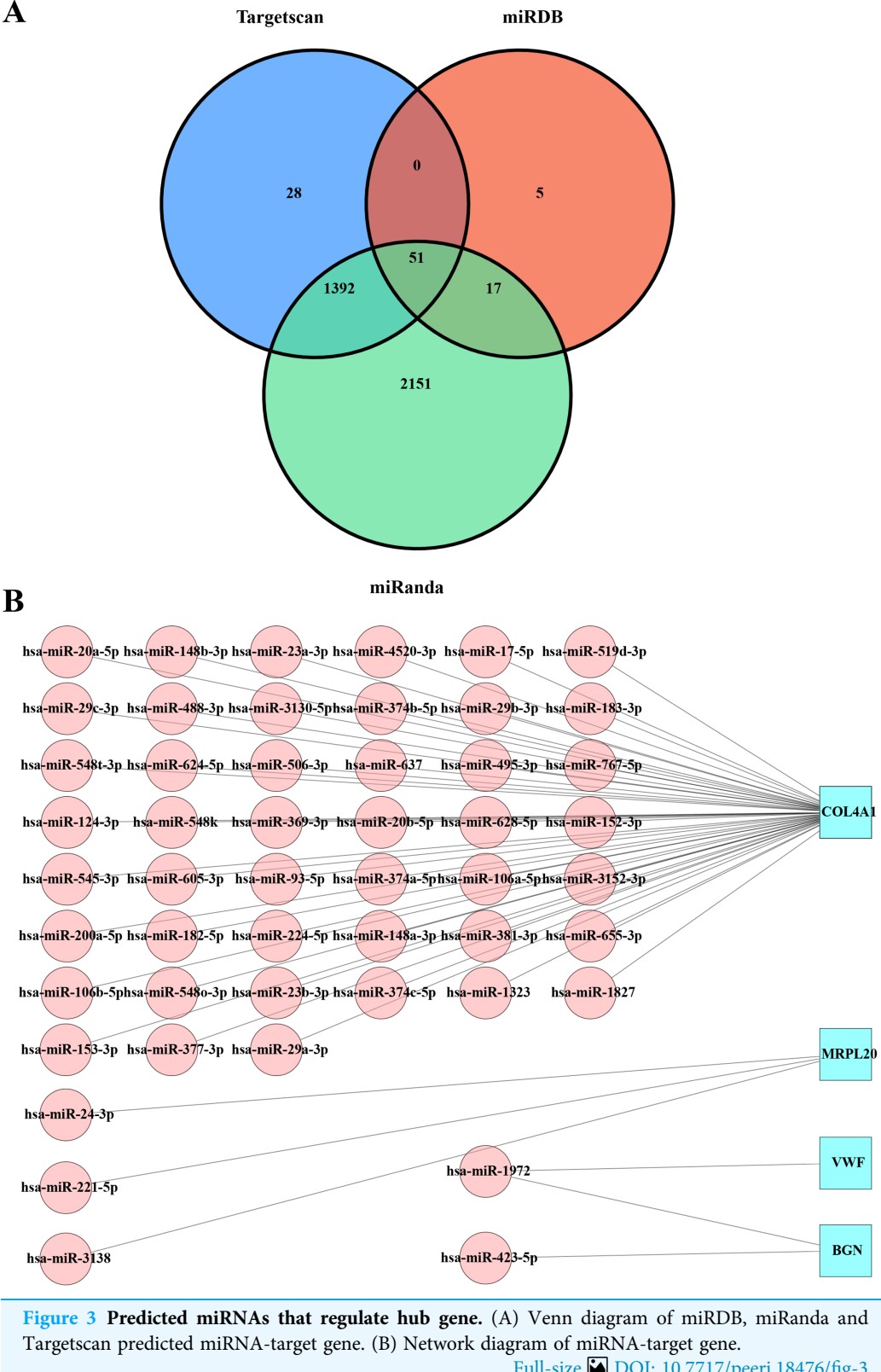

**Figure 3 Predicted miRNAs that regulate hub gene.** (A) Venn diagram of miRDB, miRanda and Targetscan predicted miRNA-target gene. (B) Network diagram of miRNA-target gene.

component in the formation of basement membrane. By interacting with other components in the basement membrane, type IV collagen α1 chain promotes stable adhesion between tumor cells and the basement membrane, thereby facilitating tumor cell migration and invasion *in vivo* (*Zhang, Wang & Ding, 2021*). In addition, VWF encoded by the VWF gene is an important plasma protein synthesized by endothelial cells and megakaryocytes, which promote vascular endothelial cell proliferation and migration to favor neointima formation and also increase platelet adhesion and aggregation by binding to the GPIb-IX-V complex on the surface of platelets (*Fan et al., 2015*). MRPL20 contributes to tumor cell migration and invasion by affecting the energy metabolism and proliferation ability of tumor cells through regulating the function of mitochondria (*Zietzer et al., 2022*). BGN encodes acetylheparin sulfate proteoglycan as a cell surface glycoprotein in the processes of cell adhesion, migration, and proliferation, and it also forms a stable adhesion structure through the interactions with other components in the basement membrane, thereby promoting the adhesion of tumor cells to the basement membrane and facilitating the migration and invasion of tumor cells *in vivo* (*Haupt et al., 2009*). It was found that hsa-miR-1972 was one of the important miRNAs that targeted VWF (Fig. 3B), indicating that there may be a potential association between hsa-miR-1972 and VWF regulation in BC progression.

## Impact of hsa-miR-1972 mimic on the gene expression and BC cell behaviors

Based on the above findings, the effects of hsa-miR-1972 mimic on the expression of several genes involved in the pathogenesis of BC and on the proliferation of MDA-MB-231 and MCF-7 cells were explored. Expression analysis showed that in MCF-7 cells, the mRNA expression of MAP2K1, MMP9, NEK1 and SHC1 was decreased following the transfection of miR-1972 mimic (Fig. 4A), while miR-1972 mimic caused the down-regulated expression of CDH1, ITGB8, MMP9, NEK1 and SHC1 in MDA-MB-231 cells (Fig. 4B). Hsa-miR-1972 had the ability to target and regulate critical genes involved in BC progression and metastasis. In addition, migration and proliferation assays demonstrated that hsa-miR-1972 mimics noticeably suppressed the growth and migration of MDA-MB-231 and MCF-7 cell lines over time, demonstrating a potential therapeutic value in suppressing BC cell proliferation (Figs. 5A–5F).

## Effects of hsa-miR-1972 mimic on the expression of angiogenic factors and VWF regulation in BC cells

It was observed that hsa-miR-1972 mimics effectively downregulated the expression of angiogenic markers, including VEGF, bFGF, TGF-β, and HIF-1α, suggesting a potent inhibitory effect on angiogenesis-related pathways (Figs. 6A–6H). In addition, Western blot confirmed a lowered protein expression of VWF in BC cells when treated with hsa-miR-1972 mimics, highlighting a direct regulatory effect of hsa-miR-1972 on VWF (Figs. 7A, 7B). The specificity of the interaction was verified by luciferase reporter gene assay as luciferase activity was significantly reduced in the presence of the hsa-miR-1972 mimic in the VWF-WT construct but not in the mutant construct (Fig. 7C). Furthermore, the

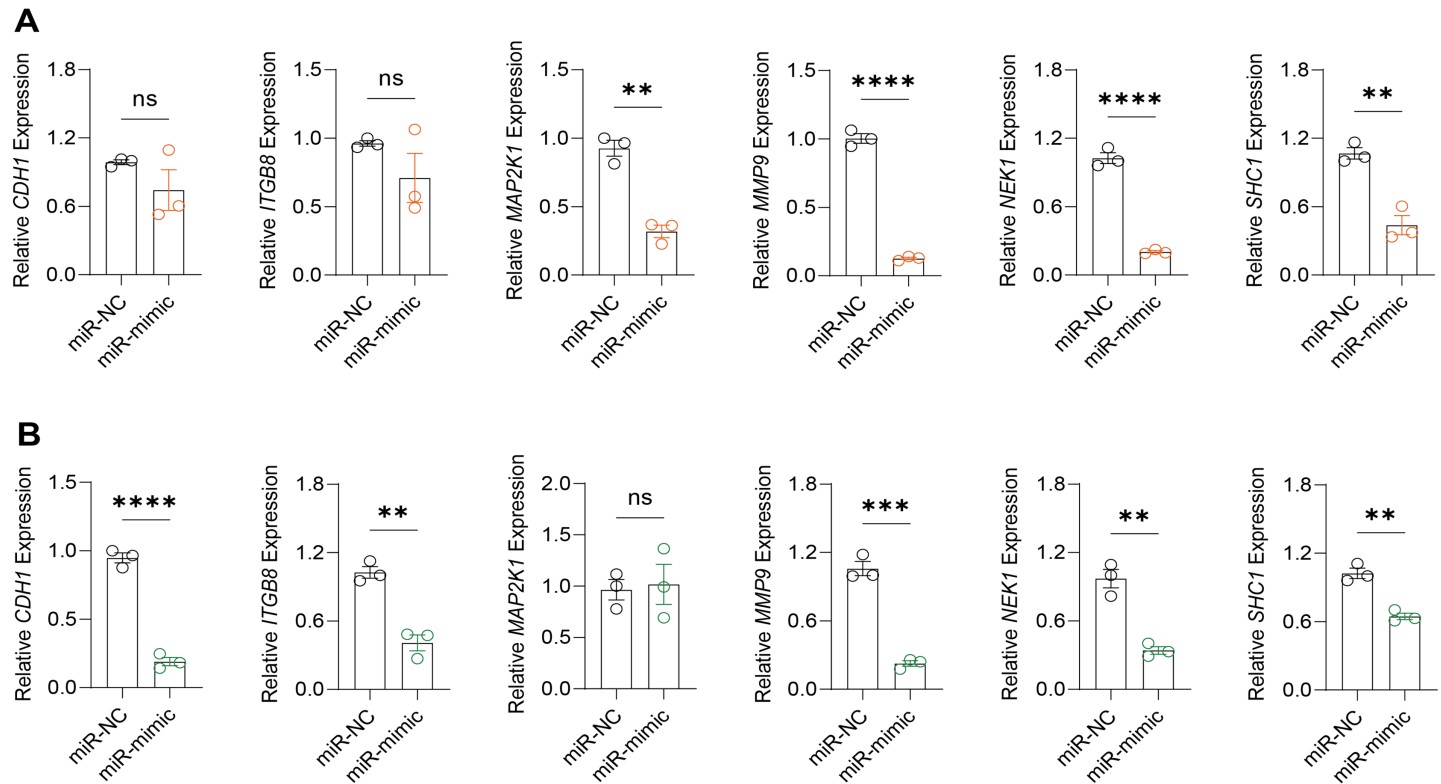

**Figure 4 Impact of hsa-miR-1972 mimic on gene expression in breast cancer cell.** (A) Gene expression levels in MCF-7 cell lines transfected with miR-1972 mimic or NC. (B) Gene expression levels in MDA-MB-231 cell lines transfected with miR-1972 mimic or NC. ****$p < 0.0001$; ***$p < 0.001$; **$p < 0.01$; ns means not significant.

significant reduction of VWF expression in MDA-MB-231 and MCF-7 cell lines suggested the potential of miRNAs as therapeutic agents to inhibit VWF-mediated pathways in BC (Figs. 7D, 7E). Collectively, these findings elucidated the role of hsa-miR-1972 mimics in regulating angiogenesis and VWF, providing novel insights for their application in BC therapeutic strategies.

## Modulation of the gene expression and the levels of angiogenic factors by manipulating VWF in BC cells treated with miRNA mimic

The effect of VWF interacting with hsa-miR-1972 on BC progression was investigated by constructing plasmids to inhibit and overexpress VWF and transfecting them into BC cell lines overexpressing hsa-miR-1972. The results showed that in cells overexpressing hsa-miR-1972 with inhibited VWF expression, the expression of tumor malignancy indicators was significantly suppressed. Specifically, all the six indicators of malignancy in the two BC cell lines were significantly downregulated after inhibiting VWF expression. However, five malignancy indicators were re-expressed when VWF was also overexpressed after overexpressing hsa-miR-1972 in the MCF-7 cell line, while four of them were upregulated in the MDA-MB-231 cell line (Figs. 8A, 8B).

Similarly, changes in the levels of pro-angiogenic factors were consistent with changes in malignant indicators. Under the conditions of overexpression of hsa-miR-1972 and

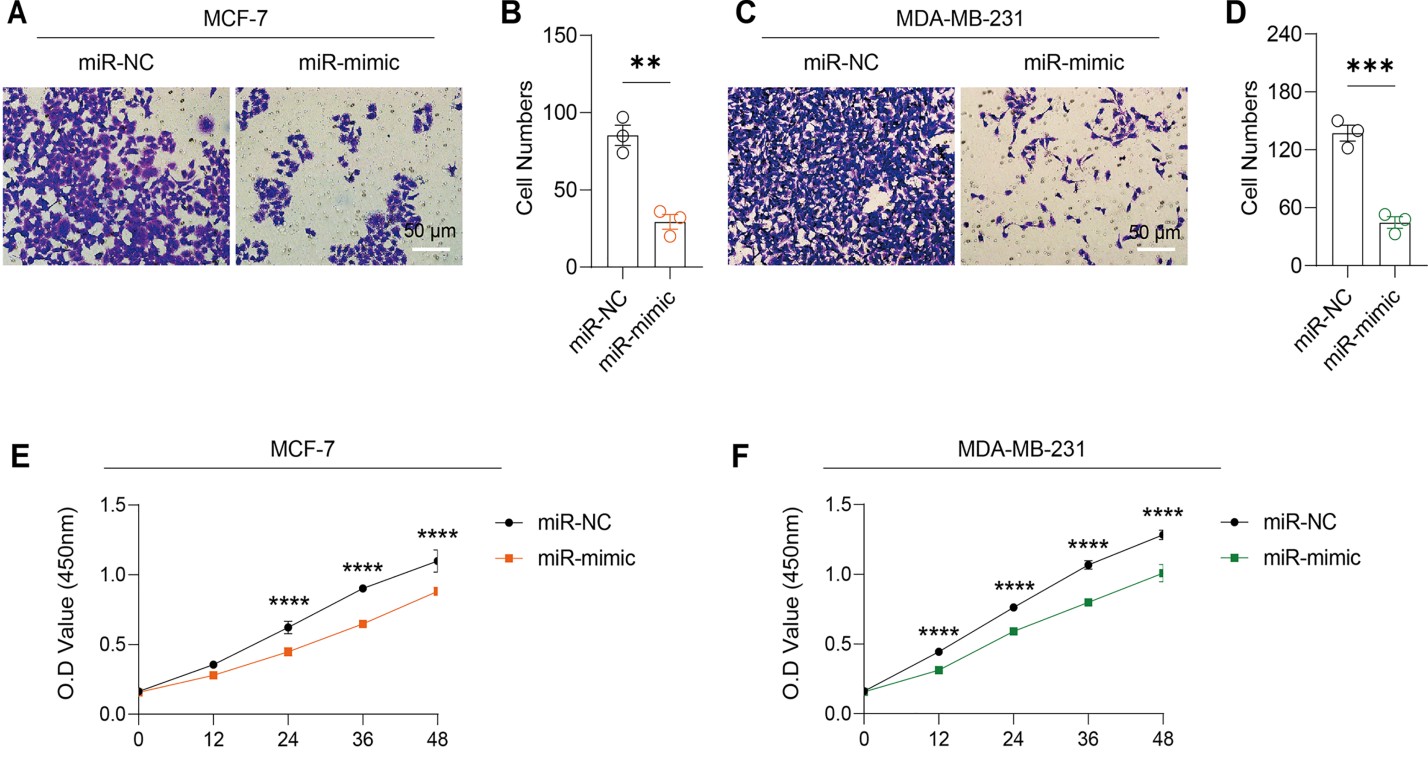

**Figure 5 Impact of hsa-miR-1972 mimic on migration and proliferation of breast cancer cell.** (A–B) Migration results of MCF-7 cell lines before and after the transfection of hsa-miR-1972 mimic. (C–D) Migration results of MDA-MB-231 cell lines before and after the transfection of hsa-miR-1972 mimic. (E–F) Cell proliferation assay of MCF-7 and MDA-MB-231 cell lines. ****$p < 0.0001$; ***$p < 0.001$; **$p < 0.01$. Samples $n = 3$, Biological replicates = 3.

inhibition of VWF expression, the concentration of pro-angiogenic factors was significantly reduced, whereas the expression of these pro-angiogenic factors was significantly elevated when VWF expression was enhanced (Figs. 9A, 9B). The results from Western blot further verified that the VWF expression levels were successfully regulated (Figs. 9C, 9D). Moreover, the results of Transwell assay also supported the above findings as inhibition of VWF expression significantly suppressed the migratory ability of BC cell lines and overexpression of VWF restored the cellular migratory ability inhibited by hsa-miR-1972 (Figs. 9E–9G). In conclusion, these data showed that the interaction between VWF and hsa-miR-1972 played a critical part in BC development, and that its regulation may provide new targets for BC therapy.

## DISCUSSION

This study focused on exploring the cell migration-related gene sets in TCGA-BRCA to reveal important genes involved in BC progression. The important modules and three hub genes (MRPL20, COL4A1, and VWF) associated with BC migration were screened by WGCNA and PPI network analysis. High-expressed MRPL20 and low-expressed COL4A1 and VWF were related to better survival outcomes. Besides, hsa-miR-1972 mimics significantly downregulated critical genes involved in BC metastasis and angiogenesis, and effectively inhibited the proliferation of BC cell lines. Overexpressing hsa-miR-1972 with

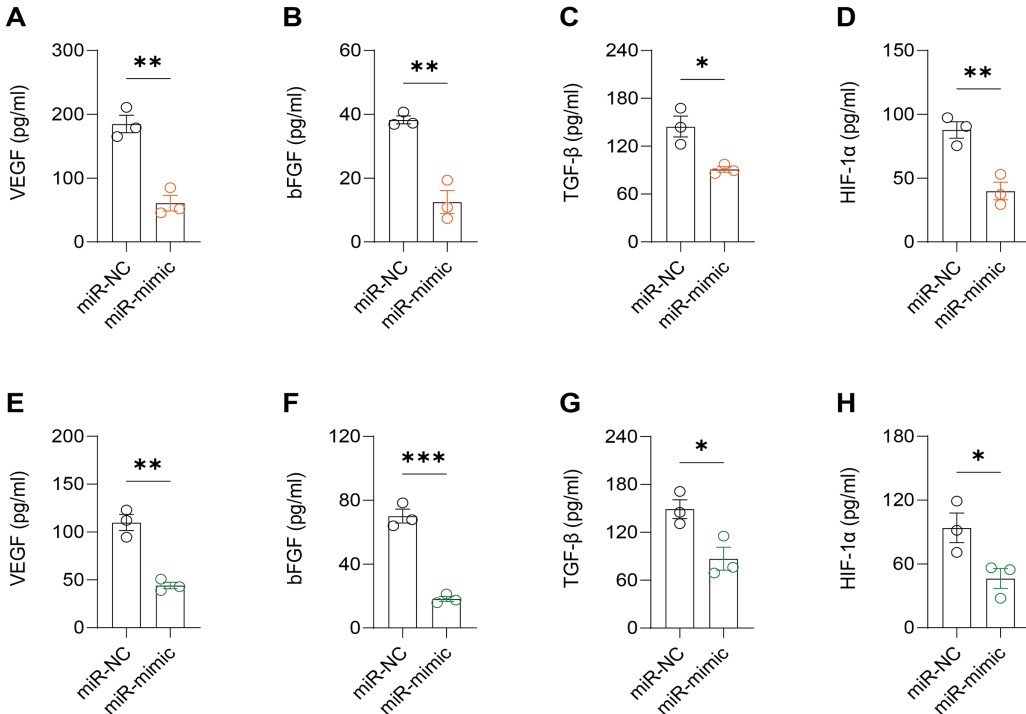

**Figure 6 Effect of hsa-miR-1972 mimic on angiogenic factor expression in breast cancer cells.** (A–D) The levels of VEGF, bFGF, TGF-β, and HIF-1α in the supernatant of MCF-7 cell culture medium before and after hsa-miR-1972 treatment. (E–H) The levels of angiogenic factors in the supernatant of MDA-MB-231 cell culture medium before and after hsa-miR-1972 treatment. ***$p < 0.001$; **$p < 0.01$; *$p < 0.05$. Samples $n = 3$, Biological replicates = 3.

inhibited VWF expression notably affected the malignant tumor markers and angiogenic factors in BC.

WGCNA has been widely used to section co-expression modules and screen hub genes for cancers, including BC. For example, one study identified differentially expressed hub genes related to immune cell recruitment in low-density protein BC, highlighting the significance of cytokine interaction pathways in this subtype. Certain hub genes are closely related to immune pathways, which also emphasizes the role of the tumor microenvironment in BC progression (*Wang et al., 2023*). By integrating WGCNA and bioinformatics analyses, a 4-gene prognostic model has shown the potential of combining gene expression data with network analyses to predict BC prognosis and immunotherapy response. Previous research identified hub genes significantly associated with the cell cycle, mitotic cell cycle processes, and cell division, providing potential biomarkers for BC (*Chen et al., 2024*). In addition, comprehensive gene expression data analysis of PPI networks in BC screened eight common central genes, namely, CCNB2, ASPM, CDK1, KIF11, CCNA2, CENPE, TOP2A, AURKB. Analysis of differentially expressed genes (DEGs) involved in various pathways including adhesion plaques and ECM-receptor interactions confirmed the significant roles of these genes in the cell cycle. Further enrichment analysis also supported the important roles of these genes in BC prognosis and diagnosis (*Elbashir et al., 2023*). Similarly, another comprehensive bioinformatics study identified pivotal

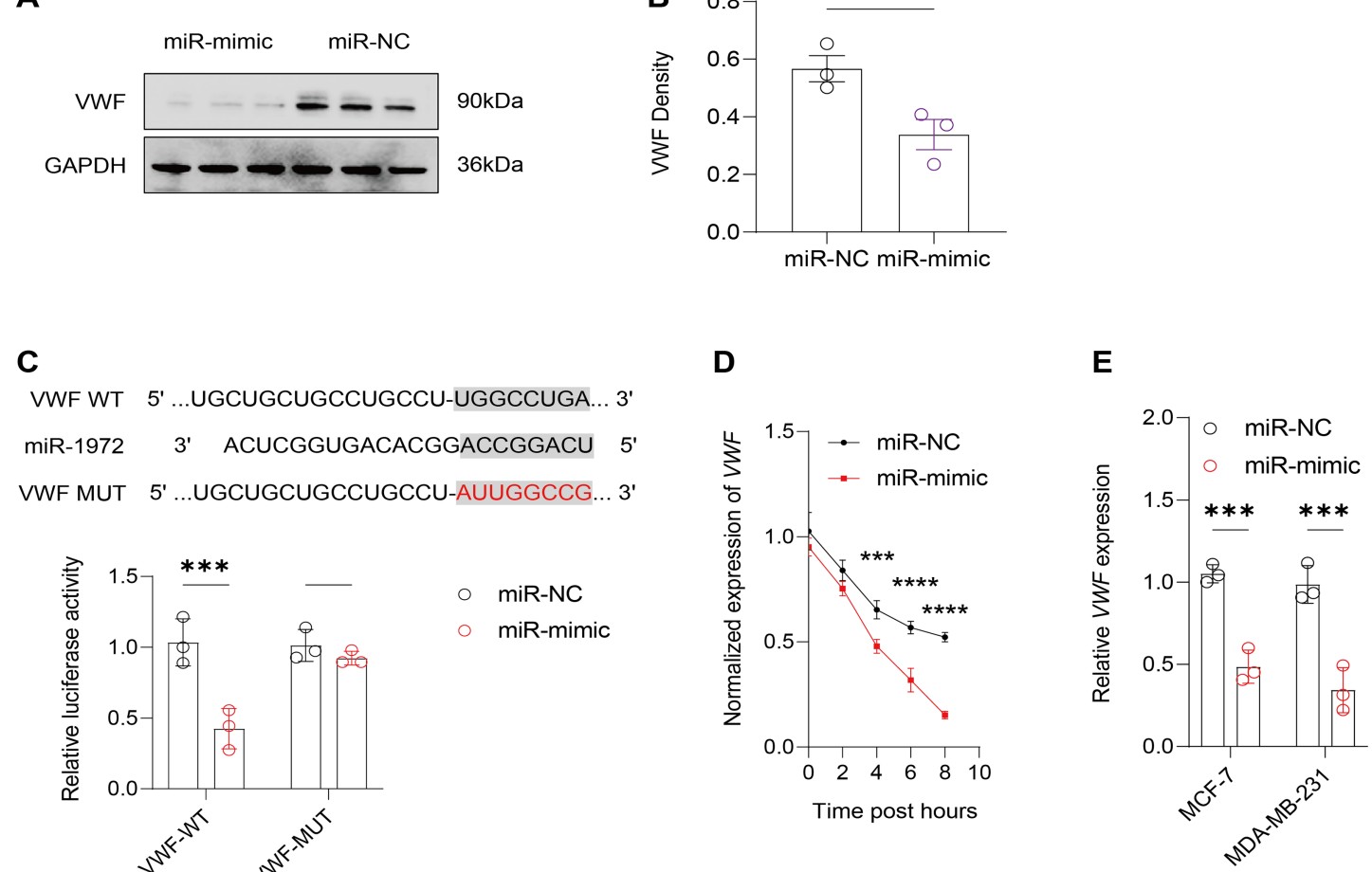

**Figure 7  Effect of hsa-miR-1972 mimic on VWF regulation in breast cancer cells.** (A) Western Blot Visualization. Using GAPDH as a loading control, VWF protein levels in miRNA mimic-treated cells were visually confirmed to be lower than NC. (B) Quantitative analysis of VWF density. (C) Schematic representation of VWF miRNA target sites. (D) Normalized VWF expression at different time points following the transfection of miR-1972 mimic/NC. (E) Relative expression of VWF in MCF-7 and MDA-MB-231 cell lines. ****$p < 0.0001$; ***$p < 0.001$; *$p < 0.05$; ns means not significant. Samples $n = 3$, Biological replicates = 3.          

genes for BC by performing KEGG pathway and GO enrichment analyses on the DEGs, and they found that cell cycle and oocyte meiotic pathways are critically involved in BC development. Moreover, this study also utilized the STRING database to assess PPI interactions, emphasizing the importance of network analysis in understanding cancer biology (*Jin et al., 2019*). To conclude, these studies all addressed the importance of using WGCNA and other network analysis tools to better understand the molecular mechanisms of BC and to identify potential targets for therapeutic intervention. The results of our analysis were consistent with these findings, as we also found that targeting the pathways and hub genes had the great potential to be explored for improving the treatment and management of BC.

Further studies revealed that the hub genes MRPL20, COL4A1, and VWF played key roles in tumor energy metabolism, cell adhesion, and angiogenesis, showing the possibility of inhibiting BC cell development and metastasis by regulating these genes and their

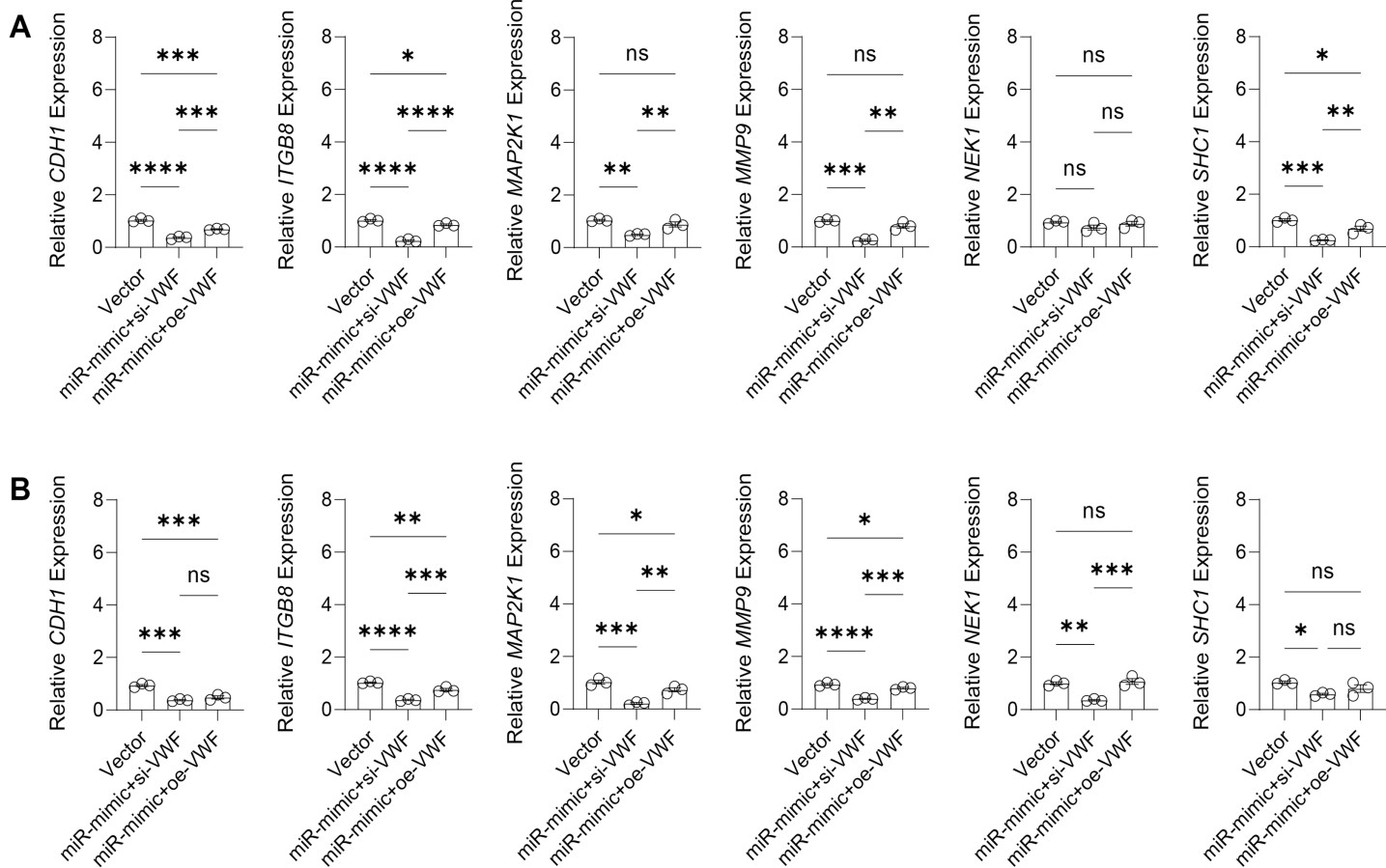

**Figure 8 Modulation of gene expression by VWF manipulation in breast cancer cells treated with miRNA mimic.** (A) MCF-7 Cell Line Malignancy Markers After hsa-miR-1972-mimic, si-VWF, and oe-VWF Treatments. (B) MDA-MB-231 Cell Line Malignancy Markers After hsa-miR-1972-mimic, si-VWF, and oe-VWF Treatments. ****$p < 0.0001$; ***$p < 0.001$; **$p < 0.01$; *$p < 0.05$; ns means not significant. Samples $n = 3$, Biological replicates = 3.

interactions. Studies have shown that genes such as COL4A1 are involved in cancer development and metastasis, suggesting the complex interplay between gene expression and cancer dynamics. A study on hepatocellular carcinoma demonstrated that COL4A1 promotes growth and metastasis through the FAK-Src signaling pathway, providing insights into the function of COL4A1 in BC (*Wang et al., 2020*). This also highlighted the importance of further investigating the potential roles of the hub genes and its miRNAs in BC prognosis and treatment. By performing cellular assays, it was found that hsa-miR-1972 significantly downregulated the expression of genes contributing to angiogenesis and cancer metastasis of BC cells, which indicated a regulatory effect of hsa-miR-1972 on the key genes in BC progression. Simultaneous regulation of VWF and hsa-miR-1972 expression could be a nuanced strategy to combat BC progression as it addressed both cell migration and angiogenic pathways. This work highlighted the complex molecular interactions in BC and the therapeutic potential of miRNAs in regulating these processes.

  This study provided a promising management direction for the diagnosis and treatment of BC based on the miRNAs and their target genes. However, the present study also has

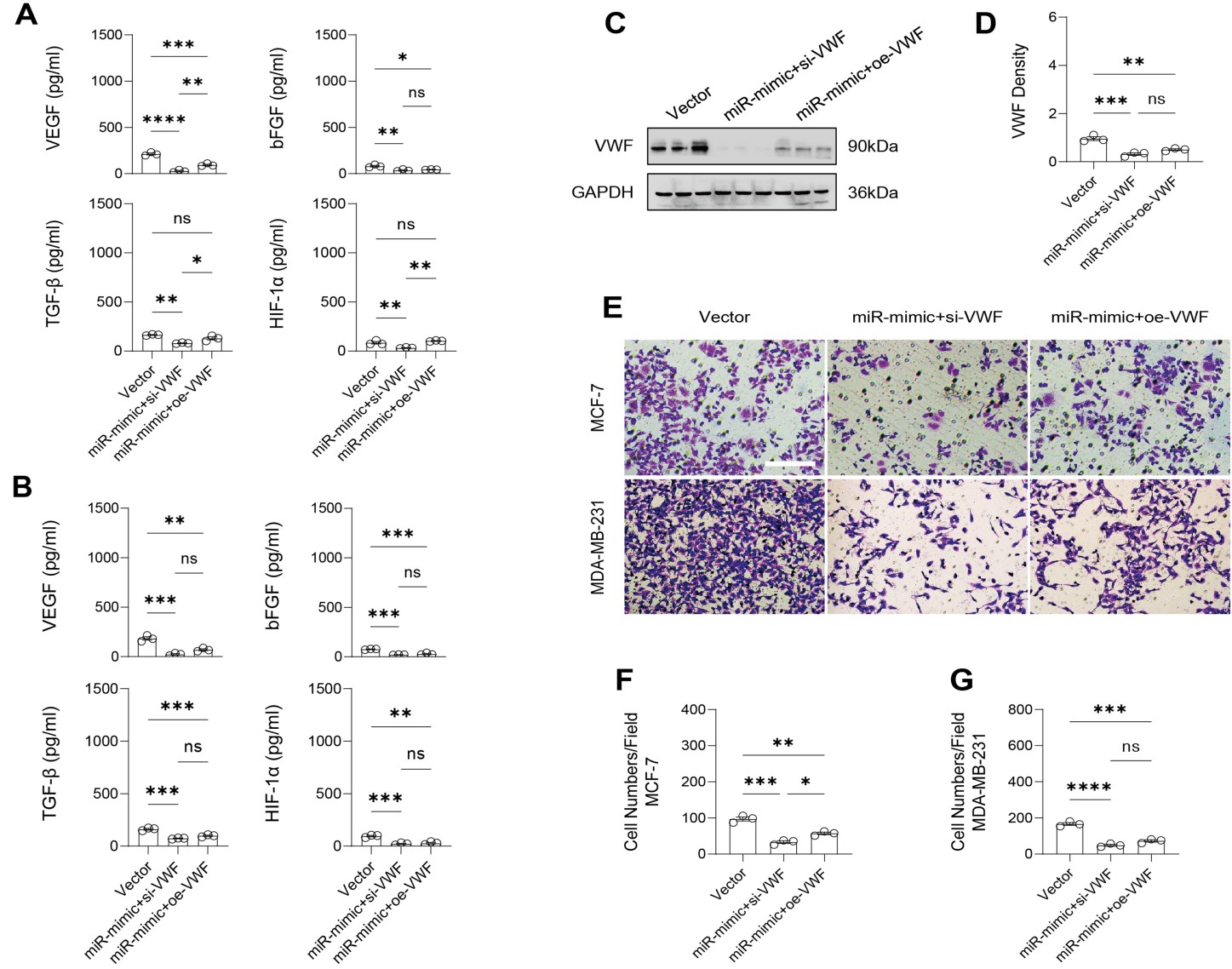

**Figure 9 Modulation of angiogenic factor levels and cellular migratory ability by VWF manipulation in breast cancer cells treated with miRNA mimic.** (A, B) Changes in Angiogenic Factor Levels in MCF-7 and MDA-MB-231 Cell Culture Supernatants After hsa-miR-1972-mimic, si-VWF, and oe-VWF Treatments. (C, D) Alterations in Intracellular VWF Protein Concentration in 293T Cells After hsa-miR-1972-mimic, si-VWF, and oe-VWF Treatments. (E–G) Migration Capacity Changes in MCF-7 and MDA-MB-231 Cell Lines After hsa-miR-1972-mimic, si-VWF, and oe-VWF Treatments, with Quantitative Cell Number Analysis. ****$p < 0.0001$; ***$p < 0.001$; **$p < 0.01$; *$p < 0.05$; ns means not significant. Samples $n = 3$, Biological replicates = 3.

some limitations. First, we only selected one of the screening miRNAs (hsa-miR-1972) to explore its role in BC progression, whether the other miRNAs could regulate BC development is anticipated to study. Second, the specific regulatory mechanism of hsa-miR-1972 in BC are needed to further explore by the help of extensive tests in the future, such as animal experiments and clinical trials.

## CONCLUSIONS

This report highlighted the potential of targeting specific miRNAs and their related hub genes as a therapeutic strategy for BC treatment, providing useful insights into the mechanisms underlying BC development.

### Funding

The authors received no funding for this work.

### Competing Interests

The authors declare that they have no competing interests.

### Author Contributions

- Changjiang Yu conceived and designed the experiments, analyzed the data, prepared figures and/or tables, and approved the final draft.
- Tao Zhang conceived and designed the experiments, analyzed the data, authored or reviewed drafts of the article, and approved the final draft.
- Fan Chen performed the experiments, analyzed the data, authored or reviewed drafts of the article, and approved the final draft.
- Zhenyang Yu performed the experiments, analyzed the data, prepared figures and/or tables, and approved the final draft.

### Data Availability

The raw data is available in GitHub and Zenodo:

- https://github.com/8ChangjiangYu/Updated-data.git

- 8ChangjiangYu. (2024). 8ChangjiangYu/Updated-data: Raw data in revision 2.0 (v.1.1.4). Zenodo. https://doi.org/10.5281/zenodo.13906988.

### Supplemental Information

Supplemental information for this article can be found online at http://dx.doi.org/10.7717/peerj.18476#supplemental-information.

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
