# Peer review of "The impact of hsa-miR-1972 on the expression of von Willebrand factor in breast cancer progression regulation"

_PeerJ, doi:10.7717/peerj.18476_

## Round 0.1 · original submission · Major Revisions

Based on reviewers' comments, we invite you to revise and resubmit your manuscript, addressing the following points:
1. Provide a more detailed introduction to hsa-miR-1972 and its role in cancer progression, particularly in breast cancer.
2. Clarify the purpose of measuring cancer migration scores and its relevance to hsa-miR-1972's impact on breast cancer progression.
3. In the introduction, include the morbidity and mortality rates of the four main molecular subtypes of breast cancer. Also, discuss whether the samples in your study distinguish these subtypes and if they have any impact on the results.
4. Clearly state the purpose of each analysis in the materials and methods section to avoid confusing the reader. Explain the connection between the sections "Prediction of miRNAs regulating the hub genes" and "Impact of hsa-miR-1972 mimic on the gene expression and BC cell behaviors."
5. Justify the direct measurement of angiogenic factors and VWF regulation factors in the context of the study's focus on cell migration characteristics and analysis.
6. In the background of the abstract, summarize the purpose of this paper. In the methods section of the abstract, include the software and R packages used in the study.
7. Discuss the prospects and limitations of your study in the discussion section. Provide a summary statement of the entire manuscript in the first paragraph of the discussion to improve coherence.
8. Deepen the description of the results in Figure 3, including hypothetical descriptions when discussing the Western blot results and elaborating on the specific regulatory role of hsa-miR-1972 on the malignant phenotype of cancer cells.
9. In Figures 4, 5, and 6, explain the range of p-values for *, **, ***, and **** in the figure legends. Improve the visibility of lines in Figure 3B. Correct the mislabeling of Figure 6 as Figure 3 in the paragraph starting on line 272.
10. In Figure 4C and 4E, consider including a control to test whether miR-mimic inhibits the growth of a non-cancer cell line to address the specificity of the growth inhibition effect.
11. Discuss the regulatory role of immune cells in breast cancer progression and explain why the correlation between prognostic and migration-related genes and the immune response was not examined in this study.
12. Elaborate on the specific mechanism by which hsa-miR-1972 modulates the favorable prognosis of breast cancer, citing relevant literature.
13. Discuss any novel strategies to address the heterogeneity of breast cancer in clinical treatment, supplementing with relevant literature.

Reviewer 1 ·

Basic reporting

no comment

Experimental design

no comment

Validity of the findings

no comment

Additional comments

The aim of this study was to identify genes associated with breast cancer (BC) migration through bioinformatics, predict upstream miRNAs of the genes, and then validate the mechanism by which upstream miRNAs target BC migration genes to regulate BC progression through combined cellular experiments. In this study, single-cell transcriptome analysis and WGCNA were performed to identify modular genes that are closely associated with BC cell migration signaling. Survival analysis and miRNA prediction of the modular genes were performed, and the regulation of angiogenic factors by miRNAs was investigated by cellular experiments. In conclusion, this is a bioinformatics combined with cellular experiments study that meets the publication requirements overall, but the following issues still need to be addressed before publication:
1. Why did this study not examine the correlation between BC prognostic and migration-related genes and the immune response to cancer? Why does it not reveal the regulatory mechanism of these genes for the level of immune cell infiltration? The regulatory role of immune cells is an important factor in the progression of BC, thus please give a reasonable explanation for this and add a description if necessary.
2. miR-1972 appears to play distinct regulatory roles in a variety of cancers, so why does it modulate the favorable prognosis of BC? Is the specific mechanism by which this miRNA regulates BC reported in the literature? Please add to this.
3. The heterogeneity of BC is an important challenge in clinical cancer treatment, and are there any novel strategies on the clinical side to treat BC heterogeneity? Please supplement the literature to improve this.
4. What is the significance of the conduct of this study? Briefly, this study elucidates the strong link between the clinical management of BC and the heterogeneity of the cancer, but why focus on the migratory phenotype of BC in the analysis? It is recommended that the introductory section elucidate reports related to BC migration and explain the necessity of focusing on genes related to cancer migration.
5. The introductory part of this study is not clear enough and elaborates more than it should, but it is unfocused, so please clarify specifically the important challenges facing BC treatment and what role miRNAs play in BC treatment.
6. Previous studies have demonstrated the critical role of miRNAs in BC development, maintenance, metastasis, and chemoresistance, but do the mechanisms by which miRNAs regulate BC include regulation of the metastatic phenotype of BC cell lines? After all, this paper focuses on genes associated with BC metastasis, thus suggesting additional refinements to this section.
7. Given that miRNAs have shown great potential in BC therapy and have spawned a variety of novel assays, it is suggested that one of these new assays be selected to be elaborated upon, thus giving more depth to the intent of this paper.
8. The description of the results in Figure 3 needs to be further deepened, e.g., by adding hypothetical descriptions when describing the results of WB, and by elaborating clearly on the specific regulatory role regarding the malignant phenotype of hsa-miR-1972 on cancer cells.
9. It is suggested in the discussion section that a summarizing statement of the whole text should be added so that it is clearly presented in the first paragraph of the discussion section, thus making the whole text more coherent.
10. It is recommended that the limitations section of this study be improved, especially to highlight the follow-up ideas of the research in this paper. In addition, a summary ending should be added to the discussion section to provide an outlook, thus making the full study more sustainable.

Reviewer 2 ·

Basic reporting

In this manuscript by Yu et. al., the authors studied the role of miRNAs and their target genes during cancer progression. They found that inhibiting the expression of VWF and overexpressing hsa-miR-1972 affected the malignant tumor markers and angiogenic factors. This manuscript is overall clear and suitable for PeerJ. Addressing the following questions will help further improve the manuscript.

1. In the figure legend of Fig 4,5,6, the authors need to explain the range of p values for *, **, ***, and ****
2. fig 3b, lines are not visible
3. In the paragraph starting in line 272, Figure 6 were mistakenly annotated as Figure 3

Experimental design

1. The authors need to give a more detailed introduction to hsa-miR-1972 before going into the results.
2. In Figure 4 C,E, the authors should include another control to test whether miR-mimic also inhibits the growth of a non-cancer cell line. This will help address the specificity of the growth inhibition effect of miR-mimic.

Validity of the findings

All underlying data have been provided. They are robust and statically sound.

Additional comments

NA

Reviewer 3 ·

Basic reporting

In this study, the author explores the potential link between the hsa-miR-1972 and von willebrand factor in the development of breast cancer. Risk signatures in breast cancer were identified through bioinformatics methods. The experimental design is rigorous. However, there are still some deficiencies in details in the manuscript. Please revise it carefully according to the comments below.
1. In the background of abstract section, the purpose of this paper should be summarized.
2. In the methods of abstract section, the software and R packages used in this article should be supplemented.
3. In the introduction, the Breast cancer (BC) was categorized into four main molecular subtypes, what are they morbidity and mortality rates.
4. In the introduction, the authors list four subtypes of breast cancer with different tumor and treatment characteristics, do the samples in this paper distinguish breast cancer subtypes. Will different subtypes have an impact on the results of this paper.

Experimental design

5. The roles of microRNAs (miRNAs) in promoting cancer progression are diversity, author listed several examples of miRNAs, such as miR-34a, miR-200, miR-200c-3p. However, this paper focuses on hsa-miR-1972, but there are no examples of hsa-miR-1972 in cancer progression.
6. What is the significance of measuring cancer migration scores, is it an important indicator of hsa-miR-1972 affecting BC progression.
7. In the materials and methods, the purpose of each analysis should be clearly stated, otherwise it will confuse the reader.
8. What is the connection between section of “Prediction of miRNAs regulating the hub genes” and “Impact of hsa-miR-1972 mimic on the gene expression and BC cell behaviors”. The appearance of hsa-miR-1972 seems to have nothing to do with the above.

Validity of the findings

9. Why the author directly measured these angiogenic factors and VWF regulation factors in line 258, the above are in the characteristics and analysis of cell migration.

Additional comments

10. What are the prospects and disadvantages of this article, please add in the discussion section.

---

## Round 0.2 · Minor Revisions

Please could the authors amend this to specify, in every data-containing figure, how many biological and technical replicates were performed?

The authors must then upload the corresponding immunoblots, cell images and RT-PCR data for ALL REPLICATES - not just the one shown in the figure.

Reviewer 1 ·

Basic reporting

The purpose of this study is to identify the genes related to migration of breast cancer (BC) through bioinformatics, predict the upstream miRNA of genes, and then verify the mechanism of upstream miRNA targeting BC migration genes to regulate BC progression through cell experiments. Specifically, the author first conducted single-cell transcriptome analysis and WGCNA to identify modular genes closely related to BC cell migration signals. Survival analysis and miRNA prediction were conducted on modular genes, and the regulation of angiogenesis factors by miRNA was studied through cell experiments. This is a conventional wet dry combination study, with a reasonable experimental design, complete logic, appropriate statistics, and a certain degree of innovation.

Experimental design

no comment

Validity of the findings

no comment

Reviewer 2 ·

Basic reporting

The authors have addressed all my questions. I recommend the manuscript to be published.

Experimental design

NA

Validity of the findings

NA

Reviewer 3 ·

Basic reporting

In this study, the authors explored the potential association between hsa-miR-1972 and von Willebrand factors in cancer development. Identifying cancer risk characteristics through bioinformatics methods. The experimental design is rigorous. They provided detailed responses to the reviewer's questions, and the manuscript has been improved to meet the publishing requirements. I suggest publishing it.

Experimental design

no comment

Validity of the findings

no comment

---

## Round 0.3 · Minor Revisions

The Section Editor has provided the following feedback which must be addressed correctly:

We specifically stated that we wanted details in the figure legends. A non-specific statement in the statistics section is not acceptable. Again we ask: specify how many technical and biological replicates were performed for EVERY experiment shown in figures 4, 5 and 6. Two numbers for each experiment are required.

We asked for all replicate data to be uploaded. It has not been, but we are directed to a data repository site. This does not have all the immunoblot data, so far as we can see, nor does it have a sensible number of image files for the cell migration work - again only the presented files are shown.

This data is essential and should be uploaded directly to the PeerJ site so that it is available for careful inspection. Just showing a single replicate is not sufficient.

---

## Round 0.4 · accepted · Accept

I find authors have addressed all my requirements. They added statements to the figure legends, and uploaded original data for Figure 4-9. Many thanks.